

# Effects of data count and image scaling on Deep Learning training

Daisuke Hirahara[1], Eichi Takaya[2], Taro Takahara[3] and Takuya Ueda[4]

[1] Department of AI Research Lab, Harada Academy, Kagoshima, Kagoshima, Japan
[2] School of Science for Open and Environmental Systems, Graduate School of Science and Technology, Keio University, Yokohama, Kanagawa, Japan
[3] Department of Biological Engineering, School of Engineering, Tokai University, Isehara, Kanagawa, Japan
[4] Department of Clinical Imaging, Graduate School of Medicine, Tohoku University, Sendai, Japan

## ABSTRACT

**Background**. Deep learning using convolutional neural networks (CNN) has achieved significant results in various fields that use images. Deep learning can automatically extract features from data, and CNN extracts image features by convolution processing. We assumed that increasing the image size using interpolation methods would result in an effective feature extraction. To investigate how interpolation methods change as the number of data increases, we examined and compared the effectiveness of data augmentation by inversion or rotation with image augmentation by interpolation when the image data for training were small. Further, we clarified whether image augmentation by interpolation was useful for CNN training. To examine the usefulness of interpolation methods in medical images, we used a Gender01 data set, which is a sex classification data set, on chest radiographs. For comparison of image enlargement using an interpolation method with data augmentation by inversion and rotation, we examined the results of two- and four-fold enlargement using a Bilinear method.
**Results**. The average classification accuracy improved by expanding the image size using the interpolation method. The biggest improvement was noted when the number of training data was 100, and the average classification accuracy of the training model with the original data was 0.563. However, upon increasing the image size by four times using the interpolation method, the average classification accuracy significantly improved to 0.715. Compared with the data augmentation by inversion and rotation, the model trained using the Bilinear method showed an improvement in the average classification accuracy by 0.095 with 100 training data and 0.015 with 50,000 training data. Comparisons of the average classification accuracy of the chest X-ray images showed a stable and high-average classification accuracy using the interpolation method.
**Conclusion**. Training the CNN by increasing the image size using the interpolation method is a useful method. In the future, we aim to conduct additional verifications using various medical images to further clarify the reason why image size is important.

Corresponding author
Daisuke Hirahara, ffieldai@gmail.com

## INTRODUCTION

A convolutional neural network (CNN) proposed by researchers at the University of Toronto at the 2012 ImageNet Large Scale Visual Recognition Challenge (*Hijazi, Kumar & Rowen, 2015*) had significant impact on society when it achieved an approximately 10% improvement in error rate compared to previous methods. This technological development has made image classification widely known for its effectiveness, and its applications in the medical field are rapidly advancing (*Zhou et al., 2017*; *Kyono, Gilbert & Van der Schaar, 2020*; *Poplin et al., 2018*), e.g., classification of computed tomography images and mammographs, along with the prediction of cardiovascular risk from retinal fundus photographs.

By training a CNN using a large volume image data, identification can be achieved at high accuracy. However, in the medical field though, large volumes of image data are not always available. As a result, when the amount of training data is limited or only small images are available, CNNs cannot be trained as designed; thus, they cannot be used to solve problems. Generally, if the amount of data is limited when training a machine learning model for image identification, a data augmentation method is employed to mirror or rotate the available image data.

Various data augmentation methods have been proposed as of June 2020. For example, mixup (*Zhang et al., 2018*) performs augmentation by linearly complementing labels and data to create new data, Augmix (*Hendrycks et al., 2020*) realizes data augmentation by convexly joining multiple randomly sampled operations and their compositions without deviating from the original data while maintaining diversity, and random erasing data augmentation (*Zhong et al., 2020*) masks random, partial rectangular areas of an image to generate training data. These methods are effective for image classification, object detection, and person identification tasks; however, they are less effective for medical images because new data are generated and mask processing is performed. This is due to the fact that if the medical image is randomly cropped or masked, the lesion is hidden and frequently disappears in the true image.

Images of the human body have structures that originally have a fixed position of existence. For example, the liver is always on the right side of the body, and the heart is always on the left side. Therefore, even recent data augmentation methods (mixup, Augmix, and random erasing data augmentation) with right-to-left inversion or rotation may not improve the robustness of analysis for the human body images. It's worth noting that several studies are currently investigating high-accuracy identification with a small amount of data using various methods, e.g., transfer learning and multi-scale CNNs (*Bakkouri & Afdel, 2019*; *Samanta et al., 2020*). In these methods, data augmentation is performed by degrading the image with a fixed number of pixels or by degrading the high-resolution image.

However, studies using medical images often require the use of only a portion of the image. This can happen when we use CT images in the study for the lymph node. Although the original resolution of CT is enough ($512 \times 512$), the data size around the lymph nodes can only be a few tens of pixels, thereby causing low resolution. In addition, when

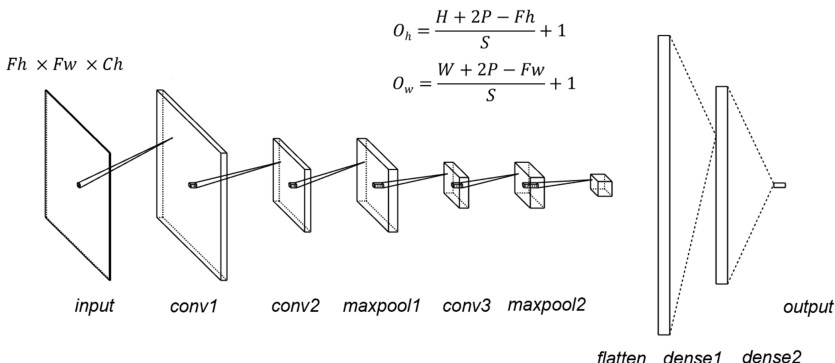

$$O_h = \frac{H + 2P - Fh}{S} + 1$$
$$O_w = \frac{W + 2P - Fw}{S} + 1$$

$Fh \times Fw \times Ch$

input    conv1    conv2    maxpool1    conv3    maxpool2    output

flatten    dense1    dense2

**Figure 1** **CNN structure.** Fh, input height, Fw, input width, Oh, output height, Ow, output width, P, padding, S, stride; kernel size = 5, stride = 1, padding = 0, dropout = 0.5.

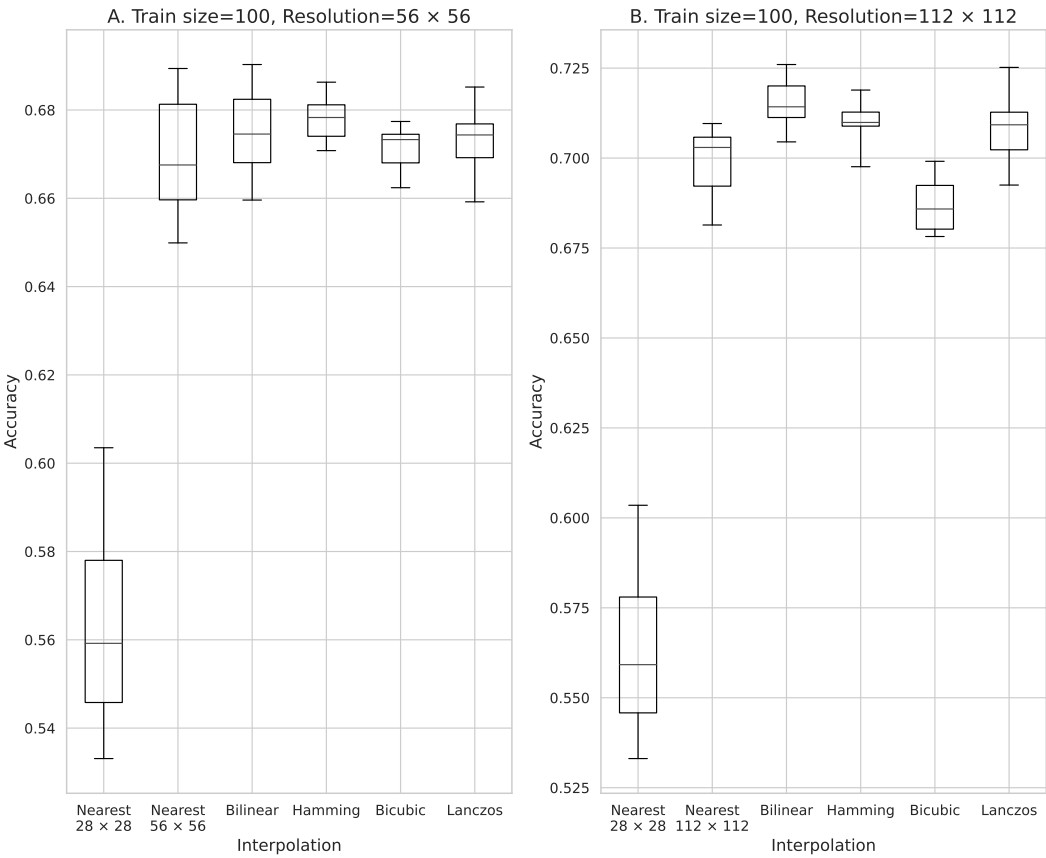

**Figure 2** **Accuracy obtained with 100 training data.** (A) 56 × 56 pixels; (B) 112 × 112 pixels.

anatomically small tissues are made into an object, an image cut out from a diagnostic image may be used. In such cases, only low-resolution image are available. Thus, in this paper, we investigated the effectiveness of using low-resolution image data processed by a pixel interpolation method as training data.

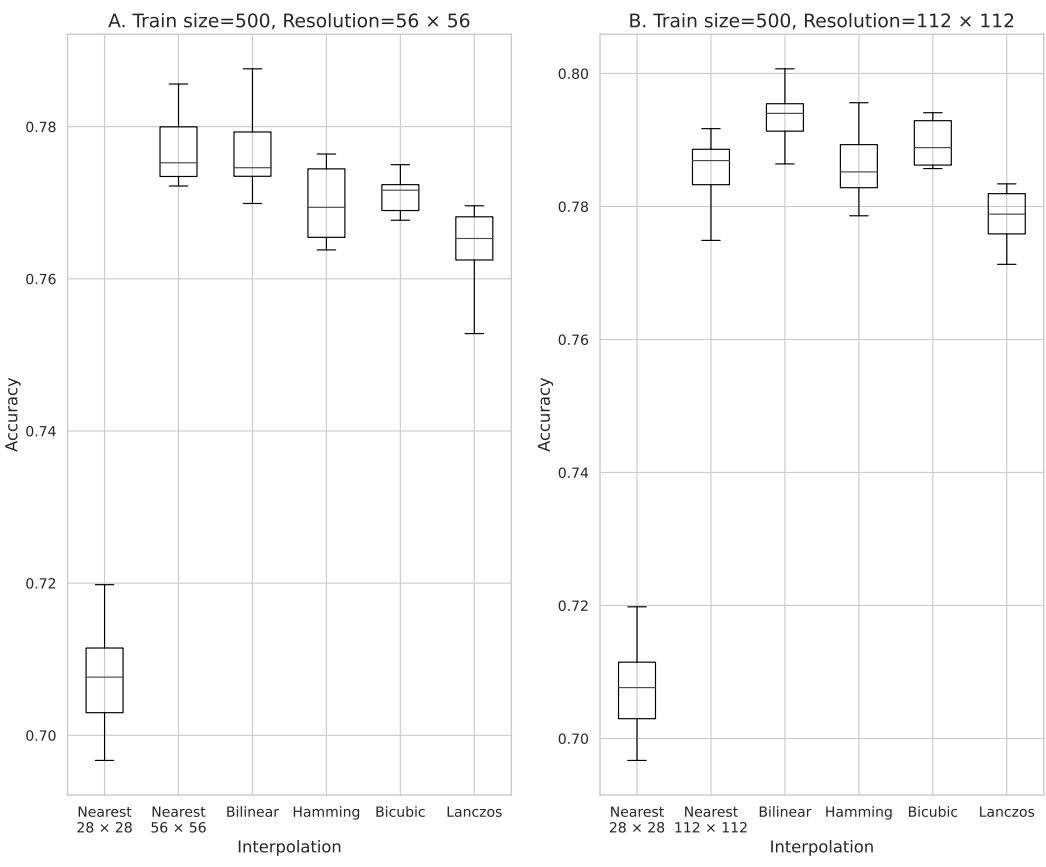

**Figure 3  Accuracy obtained with 500 training data.** (A) 56 × 56 pixels; (B) 112 × 112 pixels.

CNN is a convolution process extracting features that fit the convolution kernel. Convolution kernel sizes of 3 × 3 and 5 × 5 are commonly used. If the image data input to the CNN is small, the necessary features may not be extracted. Therefore, we increased the input image data size using the interpolation method.

In this paper, we reveal the impact of different pixel interpolation methods on model training, such as training models on low-resolution image data or training models on medical images that are cropped for the necessary part of the image.

## MATERIALS & METHODS

In this study, the Fashion-MNIST dataset (*Xiao, Rasul & Vollgraf, 2017*) was used to verify improvements in average classification accuracy. The Fashion-MNIST dataset contains 10 fashion images and is unbiased because all classes are equal. Note that monochrome images are often used in image diagnosis, and this dataset has similar features. In addition, the image size in the dataset is 28 × 28 pixels.

After examining the Fashion-MNIST dataset, we used the Gender01 data set, which predicts gender from chest radiographs published in the miniJSRT_database, as a medical

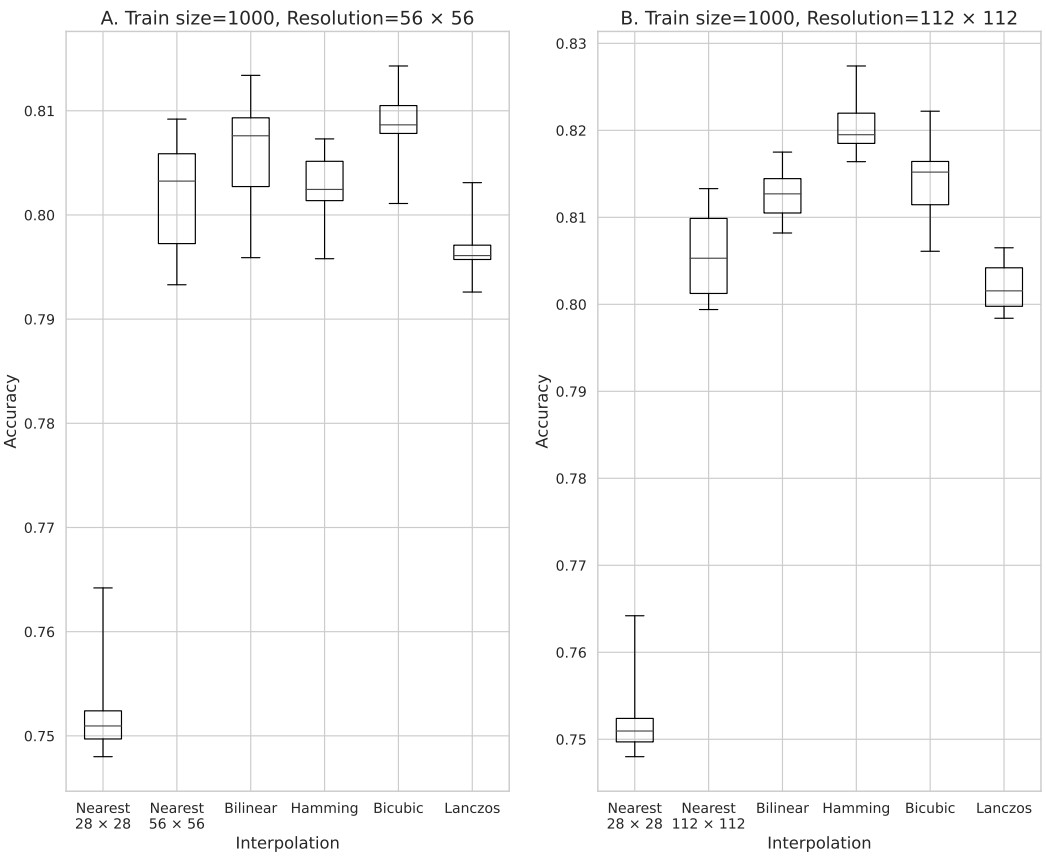

**Figure 4** **Accuracy obtained with 500 training data.** (A) 56 × 56 pixels; (B) 112 × 112 pixels.

image dataset (*Shiraishi et al., 2000*). The image size in this dataset is 256 × 256 pixels, and there are 119 and 128 images of men and women, respectively.

Moreover, Python 3.6 was used as the programming language, PyTorch (version 1.1.0) was used as the deep learning framework, and Google Colaboratory was utilized for the environment. As a deep learning model, we created and trained the CNN model. The structure of the created CNN model is shown in Fig. 1. Herein, the mini-batch method was used to train the CNN model. The training parameters included the following: the batch_size was 100, epochs were 200, Adam's method was used for optimization, and mean square error was used for loss function. We used the rectified linear unit as the activation function. The number of channels must be determined arbitrarily, and the kernel size, stride, and padding were common to all convolutional layers. Dropout was applicable to dense1 and dense2. The image data interpolation method used as input to CNN was the image processing library for Python Pillow's five pixel interpolation methods (Nearest (*Lehmann, Gonner & Spitzer, 1999*), Bilinear (*Lehmann, Gonner & Spitzer, 1999*), Hamming (*Harris, 1978*), Bicubic (*Keys, 1981*), and Lanczos (*Duchon, 1979*)). The nearest neighbor refers to and interpolates the brightness value of the pixel nearest to the reference position. In Bilinear, luminance values are linearly interpolated using 2 × 2 pixels (4 pixels)

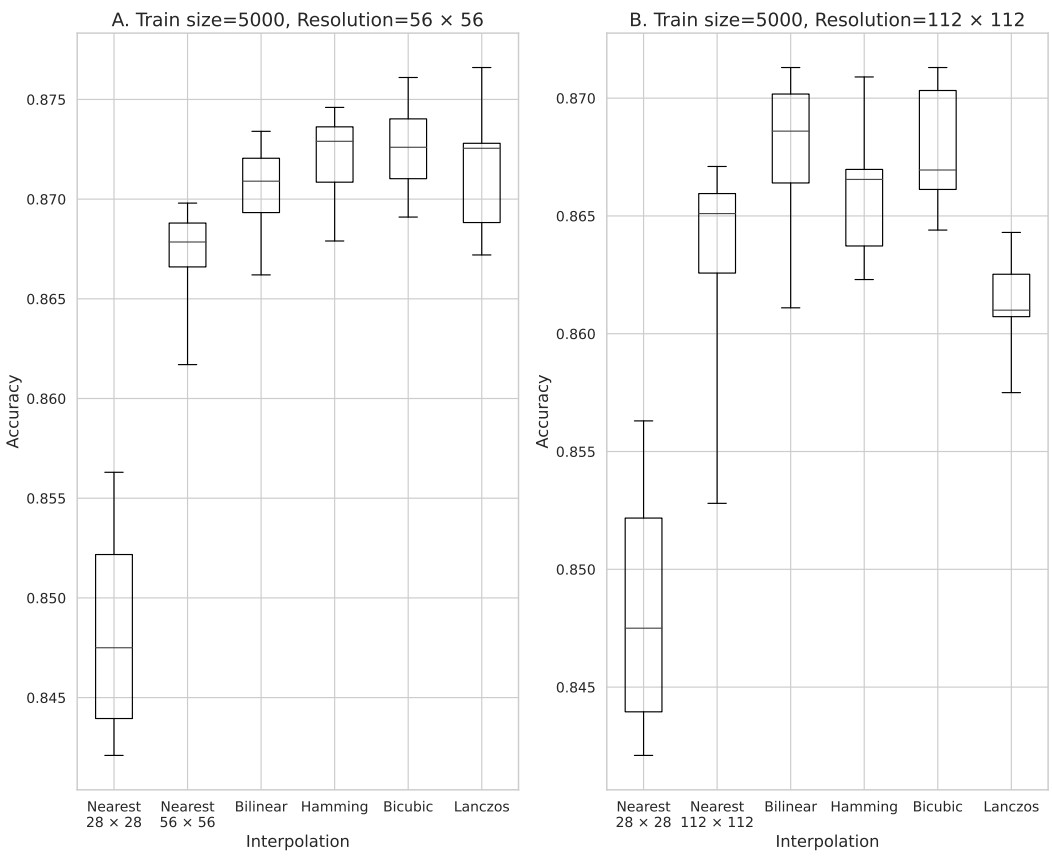

**Figure 5** **Accuracy obtained with 5,000 training data.** (A) 56 × 56 pixels; (B) 112 × 112 pixels.

in the vicinity of the position $(x, y)$ to obtain luminance values and interpolate them. In Bicubic, luminance values are obtained and interpolated by interpolating luminance values with a cubic formula using $4 \times 4$ pixels (16 pixels) around the calculated position $(x, y)$. Hamming and Lanczos are window functions, which, along with the Han window, are among the most commonly used window functions. It has a better frequency resolution and a narrower dynamic range than the Han window. Characterized by discontinuities at both ends of the interval, the Lanczos window is one of the many finite support approximations of the sinc filter. Each interpolation value is a weighted sum of two consecutive input samples. For additional details about each method, refer to Pillow's documentation and original papers (*Lehmann, Gonner & Spitzer, 1999*; *Harris, 1978*; *Keys, 1981*; *Duchon, 1979*).

In the Fashion-MNIST dataset investigation, the total number of coupling layers in the CNN was changed from 256 to 5,184 when image interpolation was doubled and 33,856 when it was quadrupled. Here, a small subset of images (100, 500, 1,000, 5,000, 10,000, and 50,000) was constructed from 60,000 images such that the number of images per class was uniform.

Then, classification accuracy was obtained by identifying 10,000 images used as test data. The training and evaluation processed were each performed 10 times.

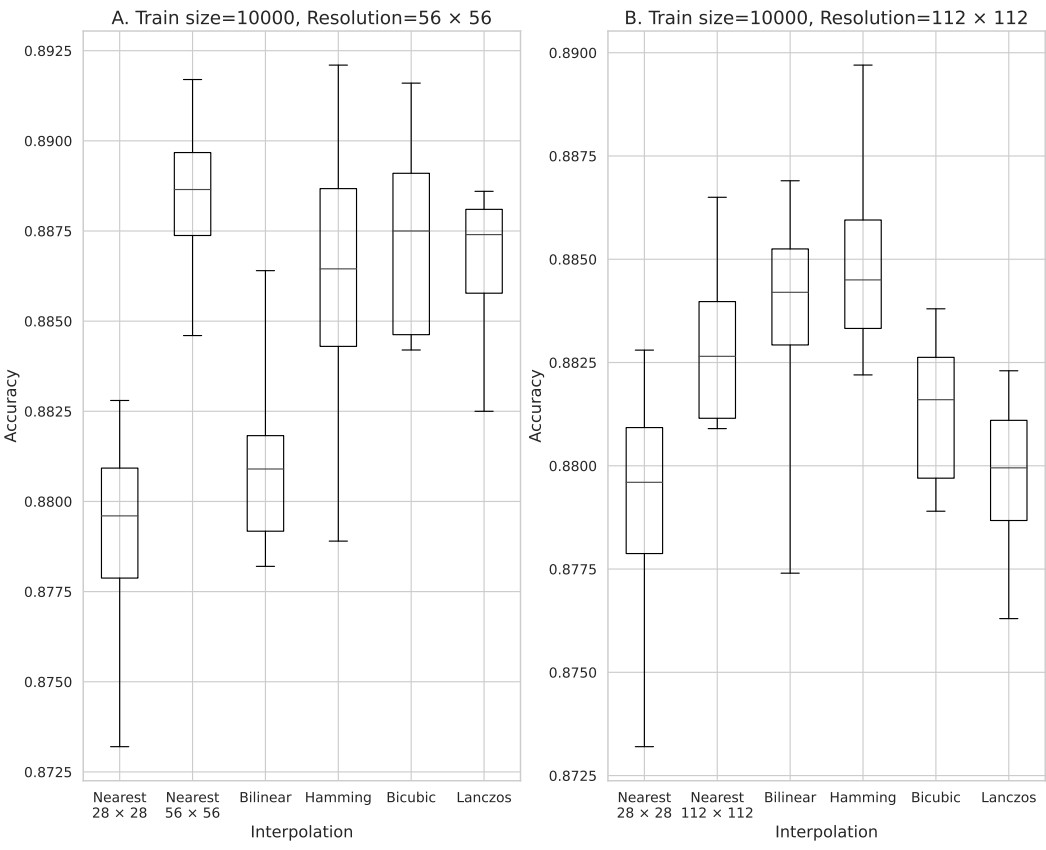

**Figure 6** **Accuracy obtained with 10,000 training data.** (A) 56 × 56 pixels; (B) 112 × 112 pixels.

In addition, it was compared with a conventional image data augmentation method, i.e., rotation and inversion. Here, horizontal inversion and rotation of ±20° were applied randomly to a group of training images, and the training and evaluation processes were each performed 10 times.

For the Gender01 dataset, the image size was reduced to 28 × 28 pixels by resizing. Here, considering an image as an input to the training model, the number of fully-connected layers was changed from 256 to 5,184 on doubling the resolution using five different Pillow's pixel interpolation methods and from 256 to 33,856 on quadrupling the resolution. The Gender01 dataset was examined with ten-fold cross-validation because the total number of datasets was only 247.

## RESULTS

Figure 2 shows the results of training using 100 pieces of data processed using the five image interpolation methods. With a mean classification accuracy of 0.563 for the training model in the source image, in which the image size was not expanded by the interpolation method, the average classification accuracy was improved for all models trained with image enlargement data using the pixel interpolation method. The method that most improved the

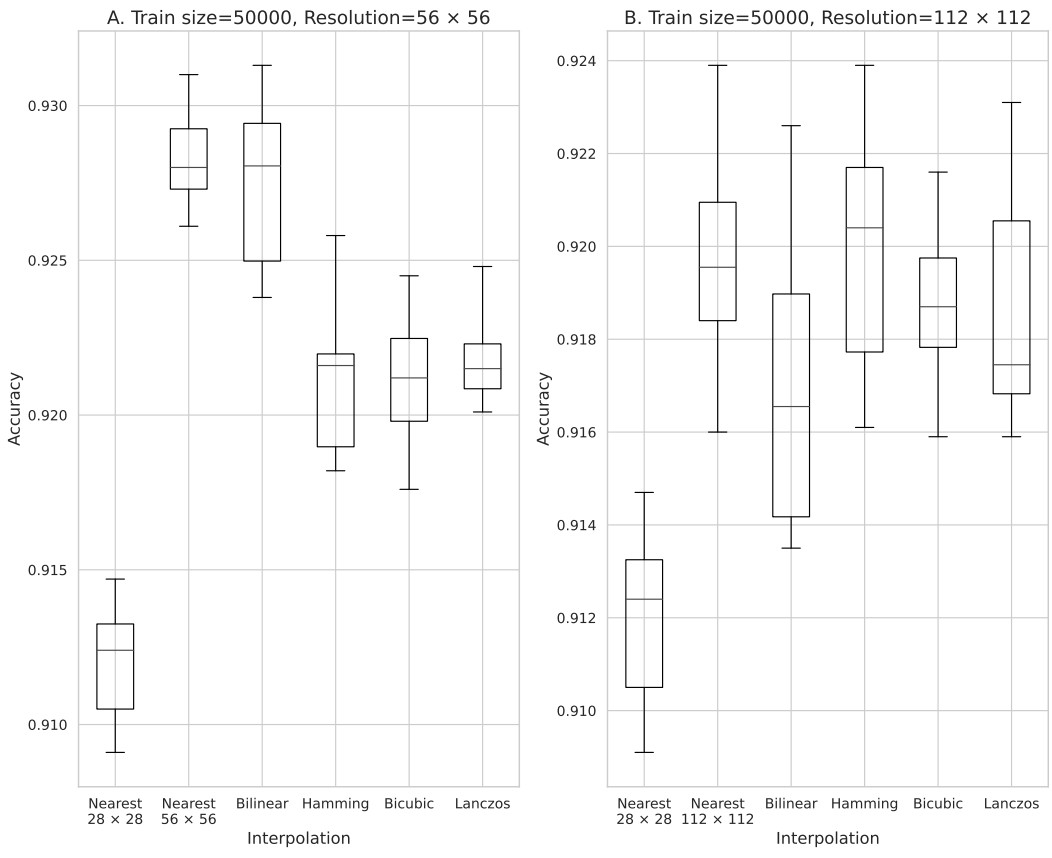

**Figure 7** **Accuracy obtained with 50,000 training data.** (A) 56 × 56 pixels; (B) 112 × 112 pixels.

classification accuracy was the two-fold magnification method with an average classification accuracy of 0.698 for the Box and the four-fold magnification method with an average classification accuracy of 0.715 for the Bilinear. The mean classification accuracy was improved by up to 0.152 over the training model for the source image.

The results obtained with data counts of 500, 1,000, 5,000, 10,000, and 50,000 are shown in Figs. 3, 4, 5, 6, and 7, respectively. In all cases, the data obtained when the image was enlarged and trained were more accurate than the original data.

Figure 8 shows the results of training by increasing the number of data by rotating and inverting the image. Here, for comparison, the results of training using the data obtained by the Bilinear image interpolation method are also shown in Fig. 8. The minimum average classification accuracy when rotating and inverting the data augmentation was 0.580 when the number of training data was 100. The maximum average classification accuracy was 0.912 when the number of training data was 50,000.

The minimum mean classification accuracy on performing data augmentation using the Bilinear image interpolation method was 0.675 for 100 training data and an image size of 56 × 56. The maximum average classification accuracy was 0.927 for 50,000 training data and an image size of 56 × 56.

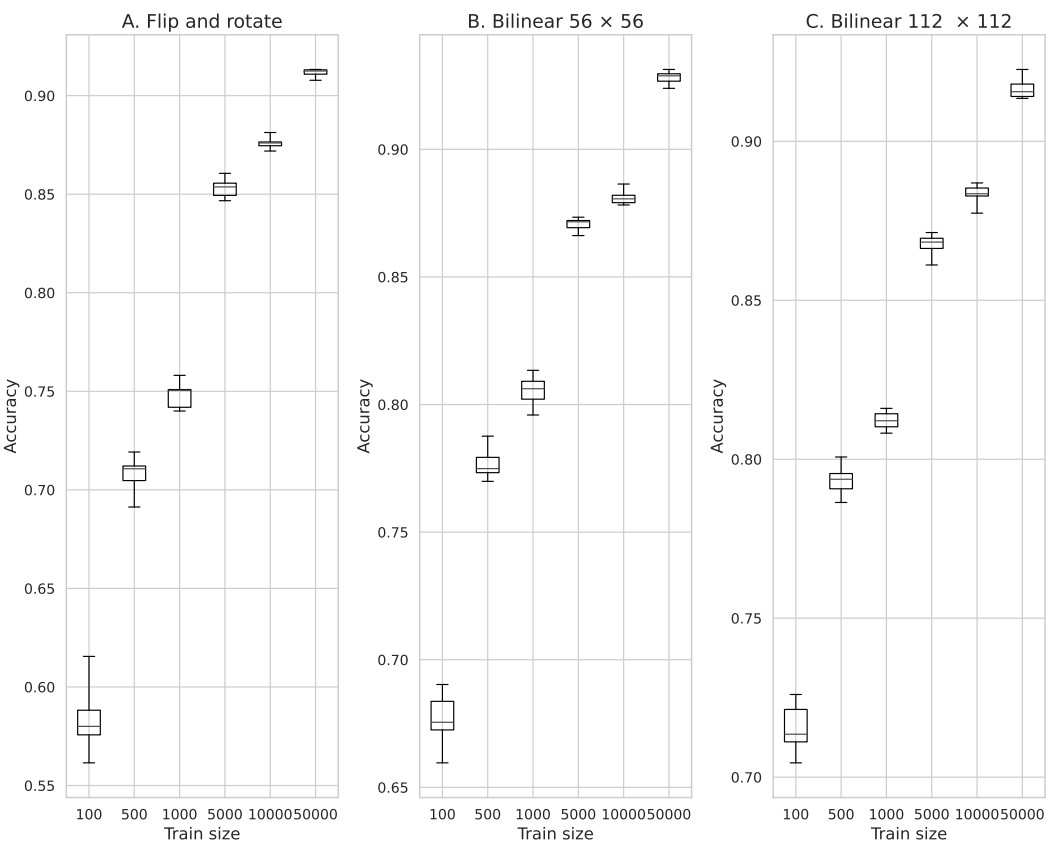

**Figure 8** **Comparison of the classification accuracy between training models of data augmentation using rotation and inversion and image augmentation using Bilinear.** (A) Data augmentation by rotation and inversion, (B) Bilinear 56 × 56, (C) Bilinear 112 × 112.

Finally, the medical image results are shown in Fig. 9. Here, the average accuracy of 28 × 28 pixels was 0.85, and the average accuracy of 56 × 56 pixels 0.863 to 0.880. By doubling interpolation, the average accuracy improved by 0.0163 to 0.023. At 112 × 112 pixels, the average accuracy was 0.875 to 0.892, which is an improvement of 0.025 to 0.042 with four-fold interpolation. At 28 × 28 pixels, the minimum accuracy was 0.625, which was 0.2 less than the average. Here, the minimum accuracy was 0.75 when the image was enlarged, which resulted in stability.

## DISCUSSION

The obtained results demonstrate that as the number of image data used for training increases, the features of images that can be extracted by CNN also increases and the effect of increasing the features obtained by image interpolation decreases. From these results, it is considered that the effect of image interpolation is high even if the number of data used for training is small.

Although the Wu et al. study is also due to color images, the true class is out of the top five predictions when models trained on 256 × 256 low-resolution images are used.

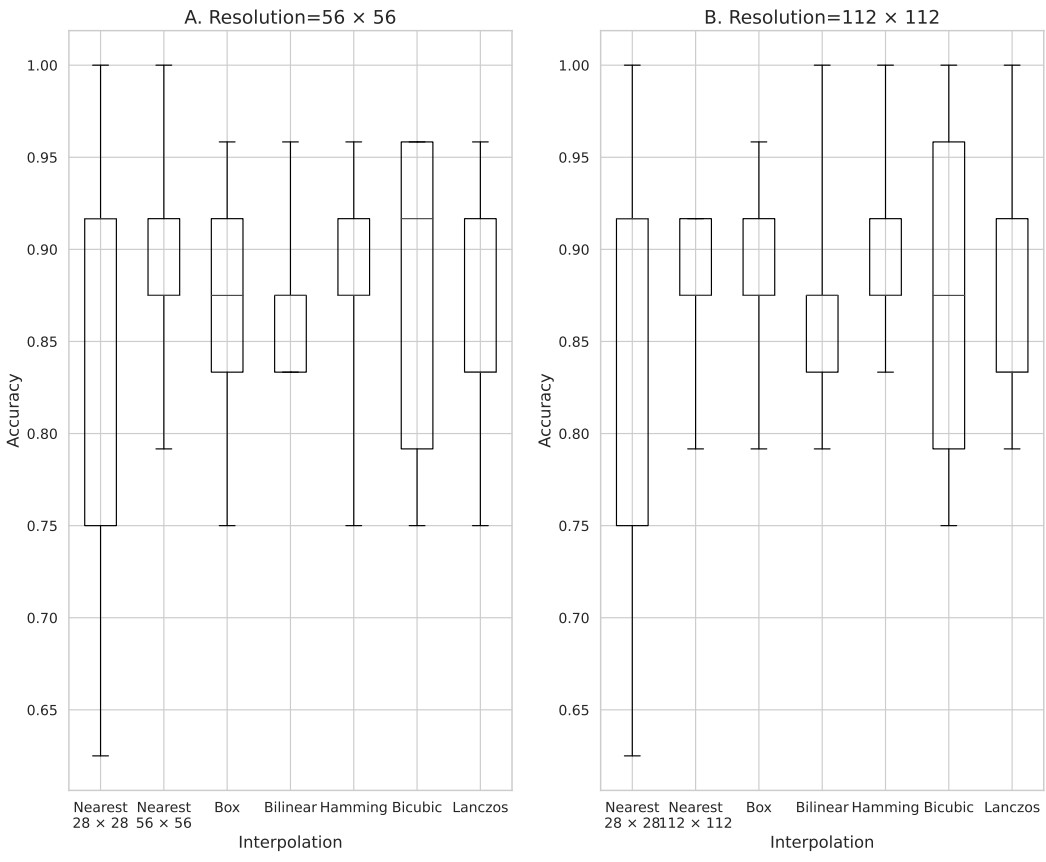

**Figure 9** **Accuracy of chest radiograph gender classification dataset.** (A) 56 × 56 pixels; (B) 112 × 112 pixels.

High-resolution training at $512 \times 512$ captured more features of the predicted target and showed that it recognized images in second place (*Wu et al., 2015*).

Moreover, by comparing two- and four-fold interpolations, we found that high accuracy was obtained with four-fold interpolation even if the number of data is less than 1,000. However, if the number of data exceeded 5,000, two-fold interpolation provided higher accuracy than the four-fold interpolation. Although image data interpolation increased the feature value, effective information was not always generated by the interpolation process. Thus, to improve the accuracy further, we consider that it is insufficient to only increase the feature value. In other words, the quality of the data must also be improved. This result demonstrates that when the number of data is large, unimportant information is included, which results in an inverse effect because many feature values are increased by the algorithm in the four-fold interpolation.

From the results obtained using data augmentation by image inversion, image rotation, and image interpolation, we found that when there are many normal images in the original image data, the information created by these procedures does not function as valid data. Thus, even though the number of data increases, the increased amount of data does not

result in improved accuracy. In addition, the accuracy improvement obtained using image interpolation was remarkable when the number of data was small; thus, we consider that image interpolation is an effective method to improve accuracy compared with the conventional method, i.e., rotation and inversion.

## CONCLUSIONS

In this paper, we investigated the effect of using interpolated image sizes for training data on the classification accuracy using five image interpolation methods on monochrome and low-quality fashion image data. For all methods, we confirmed that image interpolation combined with interpolation improved the accuracy and demonstrated that this approach was particularly effective with small amounts of data. For example, when the number of data was small, four-fold interpolation was effective, however, as the number data increased, two-fold interpolation demonstrated higher accuracy. Furthermore, image interpolation was more accurate than data augmentation by rotation and inversion operations of the conventional method. Thus, even though there is an optimal value for the increased image size, it can be considered that image interpolation is a more useful preprocessing technology than rotation and inversion operations. We expect that these results will have practical implications in image preprocessing technology in the medical field, where only a small amount of low-resolution data can be obtained.

The proposed method is a preprocessing method that can be used by medical specialists without requiring machine learning technology. In addition, image classification can be further improved by utilizing the expertise on images. Finally, we expect that the proposed method will contribute to the development of medical image classification technology by fusing medical specialist expertise and easy-to-use image interpolation preprocessing technology.

### Funding
The authors received no funding for this work.

### Competing Interests
The authors declare there are no competing interests.

### Author Contributions
- Daisuke Hirahara conceived and designed the experiments, performed the experiments, analyzed the data, performed the computation work, prepared figures and/or tables, authored or reviewed drafts of the paper, and approved the final draft.
- Eichi Takaya conceived and designed the experiments, performed the computation work, authored or reviewed drafts of the paper, and approved the final draft.
- Taro Takahara and Takuya Ueda analyzed the data, authored or reviewed drafts of the paper, and approved the final draft.

## Data Availability

The code used for the study is available as a Supplemental File.

## Supplemental Information

Supplemental information for this article can be found online at http://dx.doi.org/10.7717/peerj-cs.312#supplemental-information.

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
