# Peer review of "Effects of data count and image scaling on Deep Learning training"

_PeerJ Computer Science, doi:10.7717/peerj-cs.312_

## Round 0.1 · original submission · Minor Revisions

Please prepare a new version following the recommendations of the reviewers. You must include a letter explaining how have you addressed such recommendations.

Reviewer 1 ·

Basic reporting

This paper investigates some effects of the number of data changes for the feature extraction process in deep learning training. Their investigation is mainly based on the assumption that increasing the image size using interpolation methods would result in effective feature extraction. However, I could not find any arguments in the paper whether this assumption is valid or not. The authors said that if the image data input to the CNN is small, the necessary features may not be extracted. Therefore, they increased the input image data size using the interpolation method. It would be better the authors can give more details for their use of interpolation methods.
I found that the background and problem statement are relevant to enough references.
The paper is well organized and the style of presentation is clearly understandable for the readers.

Experimental design

This paper falls within the scope of the PJ computer and it explains the utilize method in details. However, the method of using deep learning is not much originality. I can give credit for their applications to datasets of small size.

Validity of the findings

This paper highlighted the significant usages of interpolation method for the datasets of small size. Their findings are validated by using medical images.

Additional comments

This paper contains some significant material and method for extracting image features applicable to datasets of small size. It is beneficial for analyzing image data in medicine in which the only a small size of datasets are available.

Reviewer 2 ·

Basic reporting

The goal of the work is to investigate the effects of increasing the image size by using interpolation methods. The authors used a sex classification data set (Gender01) to evaluate their study. The introduction is poor literature supported. The method needs to be extended and explained deeply. The paper is in the scope of PeerJ Computer Science. The nine images in the manuscript are low quality; authors should consider using vector graphic images such as EPS.

Experimental design

Authors have shared the source code of the experimental study through a Jupyter Notebook, which is very useful, however, I respectfully encourage commenting the source code in English to reach a bigger audience. The authors used a suitable experimental setting, and the computational cost was very high.

Validity of the findings

Results corroborated that accuracy was improved by expanding the size using the proposed method.

---

## Round 0.2 · accepted · Accept

This new version revises all the comments made by the reviewers. The article is now ready for publication.

Reviewer 1 ·

Basic reporting

I have looked at the revised version of the paper.
I did not see any major drawbacks.
So I would like to recommend the paper.

Experimental design

It is good in the revised version

Validity of the findings

It is satisfactory in revised version.

Additional comments

It is an acceptable revised version.

Reviewer 2 ·

Basic reporting

The authors have improved the explanation of the method, as well the manuscript in overall.

Experimental design

The authors included some useful comments in their source code.

Validity of the findings

Results corroborated the conclusions of the work.